# MinMax Bayesian Neural Networks and Uncorrelated Representation

## Abstract

In deep learning, Bayesian Neural Networks and Dropout techniques provide the role of robustness analysis, while the minimax method used to be a conservative choice in the traditional Bayesian field. In this paper, we formulate a minimax game between a deterministic neural network $f$ and a sampling stochastic neural network $f + r * \xi$, which is a Brownian Motion or perturbation to $f$. This requires the radius $r$ to be as large as possible with fixed performance loss and normally will improve the robustness at the cost of performance for training. Ideally, $r$ should be stable after training because Brownian Motion should be isotropic. With these, a well-trained neural network can be used for out-of-distribution detection and data similarity estimation through the fixed loss both in representation learning and supervised learning, which is easier implementation and more intuitively. Also, our simple experiments on MNIST data verify that if we want to learn the uncorrelated representation through minimax coding rate loss, $r$ will not be stable unless with enough embedding dimension without bias term or Batch normalization, and these two, especially Batch normalization will have a large impact on the robustness of the trained model. At last, we study the noise perturbation impact of different distributions.

## 1 Introduction

Nowadays, deep learning, as a data-driven method, become more and more popular and has been applied to multiple areas, such as weather forecasting Bi et al. (2022), large language models Wei et al. (2022), image classification Li et al. (2019). Most neural networks are trained with supervised learning with an end-to-end framework. However, representation learning seeks a good representation of the trained data, such as learning representation by mutual information Hjelm et al. (2018), and maximal coding reduction Yu et al. (2020) which is a nonlinear principal component analysis.

Although deep learning seems to be rather successful, robustness issues still haunt the society of deep learning Liu et al. (2018); Fawzi et al. (2017); Zheng et al. (2016); Katz et al. (2017). Bayesian Neural Networks (BNN) Kononenko (1989) and Dropout Srivastava et al. (2014) techniques are two main ways for robustness. Dropout is proposed to prevent overfitting and can be seen as an approximation of Bayesian Gal & Ghahramani (2016). BNN aims to learn a distribution of neural networks through posterior estimation and use random variables to describe the weights of neural networks and update the mean and the variance at the same time Jospin et al. (2022). Previous works have shown that BNN can quantify the uncertainty of neural networks Blundell et al. (2015), is robust to the choice of prior Izmailov et al. (2021), and is more robust to gradient attack than deterministic neural networks Carbone et al. (2020). In the previous paper of Prior Networks Malinin & Gales (2018), the authors argue that the randomness of deep learning includes model uncertainty, data uncertainty, and distributional uncertainty, and utilizes the Prior Networks to do the out-of-distribution detection. A simple baseline for image classification for deep deterministic uncertainty with MCMC Mukhoti et al. (2023). In addition, the minimax method is often thought of as a robustness help for Bayesian methods Berger (2013), and it will improve the robustness at the cost of accuracy because it considers the best case of the worst case.

The minimax method or game theory has been used in deep learning for a long time. The most well-known work is the generative adversarial networks (GAN) Creswell et al. (2018) and Variational auto-encoding (VAE) Kingma & Welling (2013), which formulate the networks as a two-player

game problem using the encoder and decoder. Previous studies using the minimax game to study the robustness of neural networks are the fault-tolerant neural networks Neti et al. (1992); Deodhare et al. (1998); Duddu et al. (2019), which view the dropout as the fault node or edges of the neural networks. Recently closed-loop transcription neural networks Dai et al. (2022; 2023), designed a new two-player game between the decoder and composition of encoder and decoder with minimax coding rate-distortion (MCR), and they can regenerate the image with fixed loss.

Inspired by the minimax works in representation level Dai et al. (2023) and Bayesian Neural Network, we applied the minimax game in BNN both in representation learning and surpervised learning. To the best of our knowledge, this is the first time to applied the minimax method in BNN. Similar works are the fault-tolerant neural networks Neti et al. (1992); Deodhare et al. (1998); Duddu et al. (2019) with two differences. One is that we use perturbation rather than fault nodes or edges. The other is that we both care about the task level and the representation level. Compared with the closed-loop transcription networks Dai et al. (2023), they introduce another deterministic neural network $g$ to obtain the closed-loop transcription networks. However, we use random sampling neural networks instead to study the robustness of deep learning.

The contribution of this paper includes 2 points. First, the experiments of MinMax BNN verify that enough embedding dimension for CNN without bias term or Batch normalization layer for uncorrelated representation learning from a robustness perspective, and Batch normalization (BN) and bias, especially BN will have large impact on the robustness of simple CNN models trained by MNIST data. Second, a well-trained neural network can use stochastic sampling neural networks to find the suitable radius to do the out-of-distribution detection and estimate the data similarity both in representation learning and supervised learning with easier implementation and more intuitive in contrast to Bayesian Neural Networks.

The paper is organized as follows: Section 2 introduces the framework of Bayesian neural networks via minimax game. Section 3 presents the experiments and results. Section 4 draws the conclusions and future work.

## 2 MinMax Bayesian Neural Networks

In this section, we first propose the minimax BNN under supervised learning as follows

$$
\min_{\boldsymbol{\mu},\boldsymbol{\rho},r} \tau(\boldsymbol{\mu},\boldsymbol{\rho},r) \doteq loss(f(X,\boldsymbol{\mu})) + loss(g(X,\boldsymbol{\mu},\boldsymbol{\rho},r))
$$
$$
\text{s.t.} \quad |pre(f(X,\boldsymbol{\mu})) - pre(h(X,\boldsymbol{\mu},\boldsymbol{\rho},r))| \geq c
\tag{1}
$$

where $X$ denotes the data, $f$ denotes the center or the mean value of minimax BNN, and $g = f + r * \xi$ denotes the sampling neural network, the $loss(f(X,\boldsymbol{\mu}))$ denote the loss by the center of the BNN; and $r$ is determined by the sampling noise and the restriction condition. If we restrict the condition to equal to $0$, then this becomes a point estimation like deterministic neural networks. Note that this formulation can easy change to minimax formulation by the Langrange method Deodhare et al. (1998) and we will directly apply them into the out-of-distribution detection for surpervised learning due to the exist of similar work Duddu et al. (2019).

Next is the MinMax BNN using minimax coding rate-distortion (MCR) Yu et al. (2020) in representation learning

$$
\min_{\boldsymbol{\rho},r} \max_{\boldsymbol{\mu}} \tau(\boldsymbol{\mu},\boldsymbol{\rho},r) \doteq \Delta R(f(X,\boldsymbol{\mu})) + \Delta R(g(X,\boldsymbol{\mu},\boldsymbol{\rho},r)) + \sum_{i=1}^{k} \Delta R(f(X,\boldsymbol{\mu}), g(X,\boldsymbol{\mu},\boldsymbol{\rho},r))
$$
$$
= \Delta R(Z(\boldsymbol{\mu})) + \Delta R(\widehat{Z}(\boldsymbol{\mu},\boldsymbol{\rho},r)) + \sum_{i=1}^{k} \Delta(R(Z(\boldsymbol{\mu}), \widehat{Z}(\boldsymbol{\mu},\boldsymbol{\rho},r)).
\tag{2}
$$

Where $X$, $f$ and $g$ denote the same case with previous condition. $\boldsymbol{\mu}$ denotes the weights for the deterministic network $f(X,\boldsymbol{\mu})$, $\boldsymbol{\rho}$ denotes the randomness or variance shape of $\xi$, and $r$ is a scaling parameter like radius determined by the loss. Combine $\boldsymbol{\mu}$, $\boldsymbol{\rho}$ and $r$, we can get the sampling neural network $h(X,\boldsymbol{\mu},\boldsymbol{\rho})$. $k$ denotes the number of classes, and $\tau(\boldsymbol{\mu},\boldsymbol{\rho},r)$ denotes the object function

using MCR, $\Delta R(f(X, \boldsymbol{\mu}))$ denotes MCR by $f$ and $\Delta(R(Z(\boldsymbol{\mu})$ denotes the MCR in the subspace $Z$. $\Delta R(g(X, \boldsymbol{\mu}, \boldsymbol{\rho}))$ presents MCR by the sampling network $g$, and $\Delta R(\widehat{Z}(\boldsymbol{\mu}, \boldsymbol{\rho}))$ for the subspace. And $\sum_{i=1}^{k} \Delta R(f(X, \boldsymbol{\mu}), g(X, \boldsymbol{\mu}, \boldsymbol{\rho}, r))$ or $\sum_{i=1}^{k} \Delta(R(Z(\boldsymbol{\mu}), \widehat{Z}(\boldsymbol{\mu}, \boldsymbol{\rho}, r))$ calculate the "distance" for $f$ and $g$. For more information, please see Dai et al. (2023). This loss is one kind of principle component analysis and should be isotropic to the Brownian Motion Yu et al. (2020). In Fig I, we provide the figure of the minimum process of radius and some experiments that the radius will become stable for simple CNN models without bias or Batch Normalization.

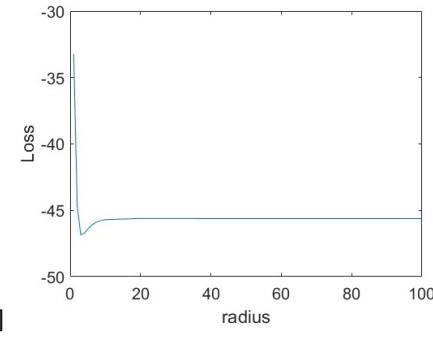 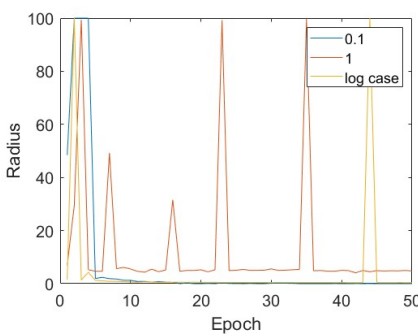

[a]         [b]

Figure 1: Radius: (a) Minimum point of radius (b) Radius during training (128 dim without bias and BN).

## 3 EXPERIMENTS AND ANALYSIS

The data sets include MNIST data LeCun (1998), Fashion MNIST (FMNIST) data Xiao et al. (2017), CIFAR-10 Krizhevsky et al. (2009), and Imagenet data Deng et al. (2009). For MNIST and FMNIST, we use the same network structure as Dai et al. (2023) and the main difference is without batch normalization layer. For CIFAR-10 data, we directly used their trained model with batch normalization Dai et al. (2023), and for Imagenet we directly used the pre-trained model VGG16. The reason is we want to test the fitness of batch normalization. What's more, we also trained the same neural network structure with supervised learning in MNIST. The optimization algorithm is Adam(0.5,0.999), the learning rate is 0.001, and normally the max epoch is 500.

Here, we use NetD to denote the discriminator $f$, NetV to denote the variance of the stochastic neural network $\xi$, and NetG to denote the sampling stochastic neural network $f + r * \xi$. To avoid negative variance, we use $\sigma = log(1 + exp(\boldsymbol{\rho}))$ in NetV to denote the standard variance, and normally we use Gaussian priors throughout the paper if without specification. The initial values of NetV are all 0, and NetD is initialized with $N(0, 0.02)$. There are two cases for the training process. For case 1 we will update $r$ via golden search for every new sampling noise $\xi$ both in maximum and minimum process, and for case 2 only update $r$ for the minimum process. Note that the way to update the variance NetV is by Bayes by Backpropagation Blundell et al. (2015) though we do not care much about NetV here. After training, we map the data to the subspace and use the knn methods Guo et al. (2003) to predict the labels implemented through scikit-learn package Kramer & Kramer (2016).

### 3.1 MAIN RESULTS

In Table I, we can see that the results of MinMax BNN are slightly worse than LDR, not even mention the results of supervised learning. Normally, supervised learning results are better than representation learning, and another reason why LDR Dai et al. (2022) performs slightly worse is it trained two neural networks with a fixed distance. Noting the main focus of this paper is about the robustness of neural network.

In figure 2, we can see that CNN without bias or BN seems to have most robustness result, and CNN with only 11 dimensions behavior very differently. CNN with bias is slight different. And CNN with Batch normalization seems breaking the robustness the largest compared with other models with 128 dimension.

Table 1: MinMax BNN results

| Models | NetD (MNIST) | LDRDai et al. (2022) (MNIST) | NetD (FMNIST) |
|--------|--------------|------------------------------|---------------|
| Case 1 | 96.28% | 97.69% | 85.82% |
| Case 2 | 96.43% | 97.69% | 85.79% |

The meaning for the minimum raduis is a fixed loss for different perturbation or Brownian motion. We already see that CNN with 11 dimensions have large width at the bottom. In Figure 3, a-d are CNN models without bias or batch normalization, and we can see that radius become stable when dimension reach 128 dimensions. If we add bias term in CNN with 128 dimensions, we can see that a slight portion of sampling radius become pretty large which means they are not sensitive for these perturbations at the cost of become more sensitive at other direction of Brownian Motion. For CNN with batch normalization, Sampling radius become more clear with many large sampling radius.

The previous paper Yu et al. (2020) claims that the rate-distortion loss should have enough dimensions to make the learned features of the subspace uncorrelated. We have shown that some condition might make the learning features is isotropic to the perturbation.

Similar results to analyze the pre-trained model VGG16 with batch normalization is given in Fig. IV.

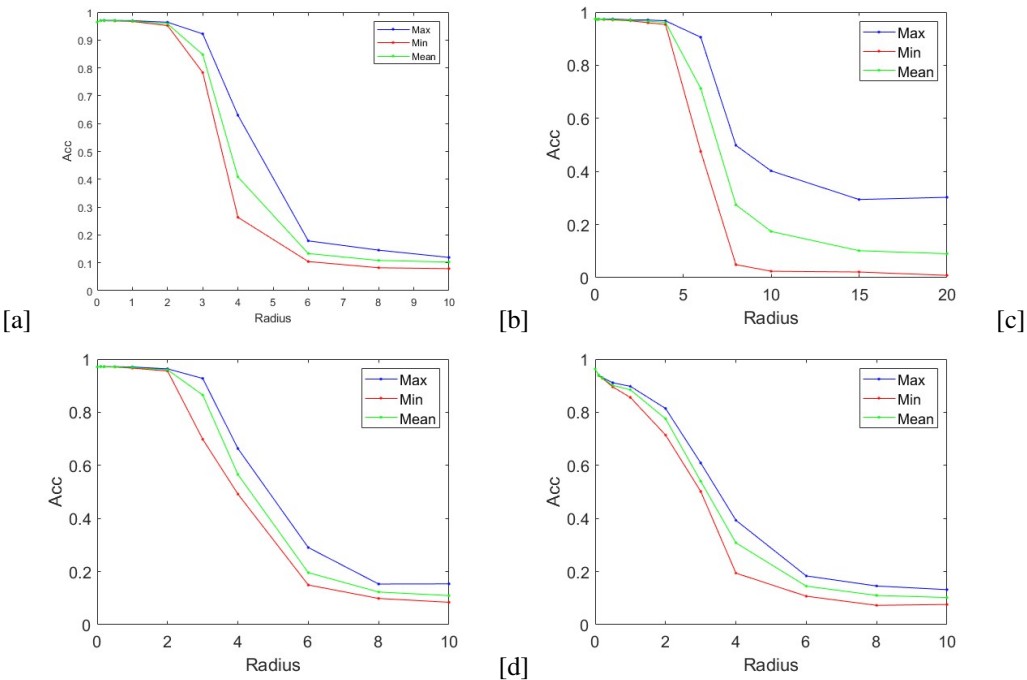

[a] [b] [c] [d]

Figure 2: Performance of different models on MNIST (a) CNN without bias or BN, 128 dim (b) CNN without bias, 11 dim (c) CNN with bias 128 dim, (d) CNN with Batch Normalization, 128 dim.

## 3.2 OUT-OF-DISTRIBUTION DETECTION

From previous results, we can see the perturbation is almost isotropic for a well-trained neural network both in well-trained supervised learning and representation learning. With these in advance, we can use this to find the maximal radius under a fixed performance to do the Out-of-distribution detection. Here, we only test the model trained by MNIST and FMNIST, Noting similar results are given in feature space with BNN and Prior Networks Malinin & Gales (2018); Mukhoti et al. (2023).

In Table 2, we can see that the radius of MNIST is about 0.5 for log case, and similar data like FMNIST is about 1.4, and cifar 10 is about 2.6, which is similar as the Gaussian noise. Similar

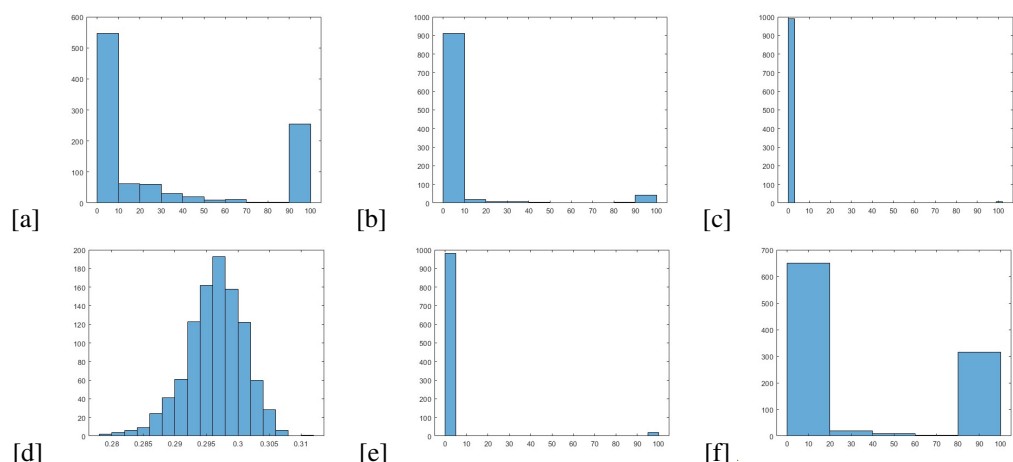

Figure 3: Histogram of trained models: (a) 11 dim (b) 32 dim, (c) 64 dim (d) 128 dim (e) 128 dim with bias (f) 128 dim with Batch Normalization.

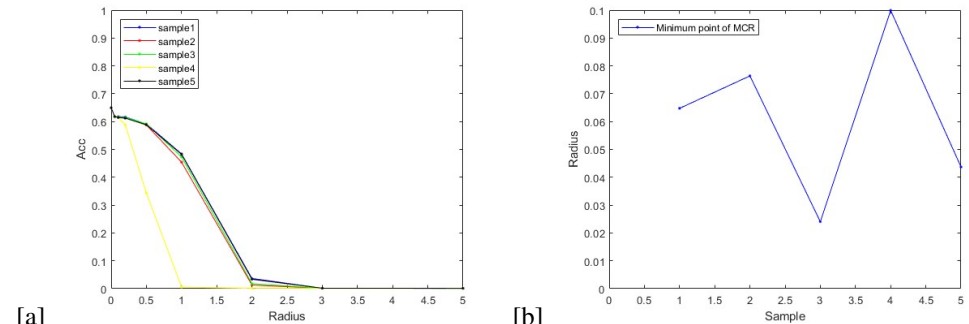

Figure 4: Radius: (a) Perturbation of VGG16 (b) Corresponding radius.

Table 2: Different r for other data sets, trained by MNIST

| r (case 1, log) | Max | Min | mean | var |
|---|---|---|---|---|
| MNIST | 0.544 | 0.481 | 0.507 | 3.1e-4 |
| FMNIST | 1.768 | 1.186 | 1.473 | 0.0184 |
| CIFAR-10(channel 1) | 3.134 | 1.999 | 2.639 | 0.102 |
| Gaussian | 3.751 | 1.778 | 2.661 | 0.311 |
| Laplace | 5.637 | 2.902 | 3.800 | 0.494 |
| Cauchy | 5.807 | 4.188 | 4.942 | 0.186 |

results for FMNIST is given in Table 3. With these, we design a online training process with multiple data source. we design an online training experiment or sequential learning scenario to see whether the model $f$ can detect a suitable data set with corresponding hyperparameters $r$ and a few iteration training. We implement the data with Gaussian noise data and other data sets (MNIST or FMNIST) for 500 epochs in total. Notice in the first 20 epochs, we use the correct data to calibrate the model $f$, while in the latter experiments, we randomly select the data from MNIST, FMNIST, or Gaussian Noise data, and the radius setting is $0.7$ for $log$ case to accept the data for training. If calculating $r$ is smaller than $0.7$, then the model will accept the data and its corresponding labels. The calculation is once after sampling the noise network $\xi$. The results are shown in Table VII. However, the performance of OOD will not be so good if introducing the BN or bias term, esppecially BN. One good thing is that this will not have accept the wrong data if having suitable radius level.

Table 3: Different r for other data sets, trained by FMNIST

| r (case 1, log) | Max | Min | mean | var |
|---|---|---|---|---|
| FMNIST | 0.591 | 0.503 | 0.546 | 5.4e-4 |
| MNIST | 1.904 | 1.479 | 1.671 | 0.016 |
| CIFAR-10(channel 1) | 2.599 | 2.155 | 2.458 | 0.012 |
| Gaussian | 3.220 | 2.189 | 2.560 | 0.076 |
| Laplace | 4.129 | 2.852 | 3.487 | 0.094 |
| Cauchy | 5.851 | 4.121 | 4.723 | 0.161 |

Table 4: Online training with data rejection

| Models | TT | TF | FT | FF | Model Acc |
|---|---|---|---|---|---|
| MNIST 1 (r=0.7,$log$) | 164 | 1 | 0 | 315 | 97.07% |
| MNIST 2 (r=0.7,$log$) | 163 | 2 | 0 | 315 | 96.84% |
| FMNIST (r=0.7,$log$) | 165 | 0 | 0 | 315 | 86.13% |

### 3.3 NOISE PERTURBATION

Finally, we test how the $r$ changes if the MNIST data and FMNIST data are corrupted by some noise. In this part, we set the corrupt ratio as $0, 0.05, 0.1, 0.2, 0.3, 0.4, 0.5, 0.6, 0.7, 0.8, 0.9$ with Gaussian random noise with normalization with the data, and calculate their corresponding radius, see figure 5. One interesting result is that the smallest $r$ is not for $0$ noise while it is about $0.4$ for MNIST and $0.3$ for Fashion MNIST. This is because a small perturbation in data can be seen by having a small radius by Taylor expansion and we can detect them as long as the data is not corrupted too much. Replacing the Gaussian noise with other normalized noise will lead to similar results, except for the Cauchy distribution, which can be seen as a Levy process with heavy-tail distribution, and will always increase $r$ all the time. Furthermore, we also compared the $\xi$ belonging to Cauchy distribution and Gaussian distribution for MNIST data, see (c).

[a] 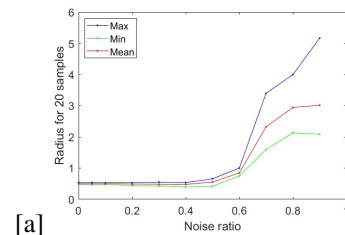 [b] 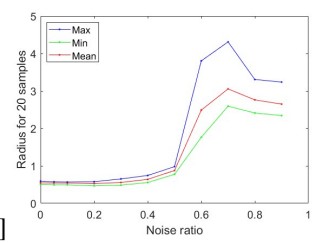 [c] 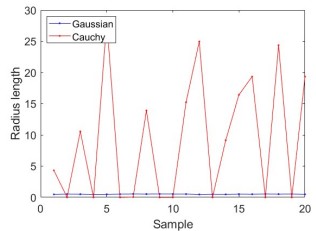

Figure 5: Noise corruption: (a) corruption in MNIST (b) corruption in FMNIST (c) Radius comparison of stochastic Neural Network with Gaussian or Cauchy noise.

## 4 CONCLUSIONS AND DISCUSSION

In this paper, we apply the minimax game to Bayesian Neural Network from the isotropic Brownian Motion point of view. With these, we verify that enough embedding dimension and non-linear activation function are needed for uncorrelated representation learning for CNN models without bias or batch normalization, and validate that batch normalization seems to influence the robust a lot in simple MNIST data set. Furthermore, a well-trained model can use the minimax BNN to do the OOD detection or estimate the data similarity, which is more intuitive and easier implementation. Last but not least, the difference between Gaussian distribution and heavy-tail distribution behavior differently, especially in the robustness of neural networks.

For future work, the first is to study how to make the learned features become more accuracy without losing robustness . The second is to study different random walks like the Levy process, and the Cauchy process, and their suitable framework in deep learning.

AUTHOR CONTRIBUTIONS

If you'd like to, you may include a section for author contributions as is done in many journals. This is optional and at the discretion of the authors.

ACKNOWLEDGMENTS

Use unnumbered third level headings for the acknowledgments. All acknowledgments, including those to funding agencies, go at the end of the paper.

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

## A  APPENDIX

