# OpenReview forum: "MinMax Bayesian Neural Networks and Uncorrelated Representation"
_ICLR.cc/2025/Conference — ICLR 2025 Conference Withdrawn Submission_

### Official Review · Reviewer_CDu8 · 2024-10-21

**Soundness:** 2
**Presentation:** 1
**Contribution:** 1
**Rating:** 1
**Confidence:** 3

**Summary:**

This work proposes a loss function inspired by minimax games and proposes that training Bayesian networks with this loss can improve robustness. The work then lists numerous numerical experiments to support its claim about the improved robustness. The experiments cover several popular datasets and test several different architectures of the networks.

**Strengths:**

- Leveraging the minimax view to improve robustness, as mentioned in this paper, is interesting. Similar idea is also seen in recent works such as [1].
- A number of numerical experiments have been listed in this work.

[1] Buening, Thomas Kleine, et al. "Minimax-bayes reinforcement learning." International Conference on Artificial Intelligence and Statistics. PMLR, 2023.

**Weaknesses:**

- First, this work is clearly neither well-written nor prepared. This includes unexplained abbreviations (e.g. LDR), numbering misalignments in multiple figures, underexplained captions (e.g. Fig 1, 4), unlabelled axes (e.g. Fig 3), and unchanged templates (page 7). Understanding the point of each of the numerical evidences is thereby difficult.
- Second, many notations are not defined. For example, I do not see the function pre() in the constraint of Eq (1) defined anywhere. And \mu, \rho, and r in Eq (1) are even defined almost 1 page later after Eq (2). Additionally, the definitions, such as "\rho denotes the randomness or variance shape of \xi", are too vague to be rigorous. Also, the summand in (2) contains nothing indexed by i.
- Also, MCR seems to be a central concept. For completeness it should be defined explicitly.
- In addition to the difficulty in following the figure explanations, it also seems to me that the authors only list metrics of the networks trained with their proposed loss. Their contribution is not demonstrated due to a lack of baseline, such as showing that networks trained with other losses or sampled deterministically are relatively not robust. The only comparison seems to be on the accuracy to LDR in Table 1, which shows a worse performance and is remarked by the authors as "not the main focus".
- The authors claim that the presence of bias and batch norm largely undermines the robustness of the trained networks, without providing further insights either intuitively or numerically.

**Questions:**

- Please see weaknesses above
- What is "golden search" for case 1?

---

### Official Review · Reviewer_kfJi · 2024-10-29

**Soundness:** 1
**Presentation:** 1
**Contribution:** 1
**Rating:** 1
**Confidence:** 5

**Summary:**

The paper reads more like a draft, and is poorly written, especially since many terms are undefined, and appear out of nowhere during the paper reading. It is difficult to even understand what contributions the authors aim for, let alone whether the experiments that they conduct actually show this.

**Strengths:**

Unfortunately, it is difficult to find any particular strength. The topic might be important, but it is not even clear what is the main motivation of the paper: Analyzing the trade-off between robustness and performance ?!

**Weaknesses:**

Unfortunately, again, I will make a generic comment: The paper is poorly written - the description of the problem, the formulation, the experiments, and the conclusions from them. Below are many comments, some are indeed minor, but they represent the general quality of the paper:

• The citation format is wrong, write, e.g. (Bi et al., 2022)

• Abstract – yellow marked text.,

• Intro, first paragraph – unclear sentence: (1) “A simple baseline for image classification for deep deterministic uncertainty with MCMC Mukhoti et al. (2023).” (2)

• Intro, line 49, it is not explained anywhere before what is meant by the “minimax method”. This makes it difficult to go through the rest of the introduction, because it is unclear what is the operation of each of the two players in the game, and what is the reward for each of them.

• First contribution – lines 69-72: It is difficult to understand, but it reads something like – “Our experiments verify that enough embedding dimension (?) has large impact on the robustness of simple CNN”. This is a vague statement. What does this concretely mean ?

• Second contribution – lines 73-76: What does it mean for a trained NN to use stochastic sampling neural networks? The claims “easier implementation” and “more intuitive” are again vague.

• Equation (1): Even though this is said to be equivalent to a minimax problem, it does not look like one. As the paper is about minimax formulation, this should be explicit.

• Equation (1): The term “pre” in the constraint is undefined, and so is *\rho* and so is *h* (this is explained only after Equation (2)).

• Line 90: What is “the restriction condition” ?

• Line 107: “*k* denote the number of classes” – Before this point, no classification task has been described.

• Lines 107-109: The term MCR and *\Delta R* are not properly explained. The role of *Z* is unclear.

• Line 113: “the figure of the minimum process of radius” is unclear.

• Line 154-157, results: It is first mentioned that “the results of MinMax BNN are slightly worse than LDR”, but then the opposite “and another reason why LDR Dai et al. (2022) performs slightly worse..”. Is NetD the result of the method proposed in the paper ? This performs worse than LDR.

• Lines 159-161: I do not see how these conclusions stem so obviously from Figure 2.

• Table I: What are each of the cases ? This is unexplained.

• Line 168: What is “Brownian motion” in this context ?

• Line 170: The radius seems to be stable already in 64 dimensions.

• Line 179: “is given in Fig. IV” – inconsistent with previous writing, should be “Figure 4”.

• Figure 4B: Why is the graph on [b] panel has this weird shape ?

• Line 232: “Histogram of trained models” – Histogram of what ?

• Tables 2 and 3: Different values of *r* are compared for different datasets, but for which method ? There is no comparison to a different method.

• Line 261: Should be “We design...”

• Line 268: “are shown in Table VII” – inconsistent with previous writing, and probably should be “Table 4”.

**Questions:**

Given the above, it is difficult to come up with intelligent or constructive questions.

---

### Official Review · Reviewer_8YPf · 2024-10-30

**Soundness:** 2
**Presentation:** 1
**Contribution:** 2
**Rating:** 3
**Confidence:** 2

**Summary:**

The study applies a minimax approach to BNNs, aiming to bolster robustness in neural networks. Using Brownian motion-based noise within a deterministic-stochastic network pairing, it analyzes how factors like embedding dimension, batch normalization, and bias affect robustness. Key results highlight that batch normalization reduces stability, whereas models with sufficient embedding dimensions are robust to perturbations. The approach’s applications in out of  detection suggest a promising direction for practical deployment.

**Strengths:**

* $\textbf{Innovative Use of Minimax in BNNs}$: Applying the minimax framework to Bayesian Neural Networks is an innovative approach that extends beyond the traditional deterministic setups seen in adversarial training or GANs.
* $\textbf{ Comprehensive Robustness Evaluation}$: The paper covers a broad range of robustness aspects, from embedding dimensions to noise impacts across various datasets, including some well-known dataset.
* $\textbf{Clear Problem Motivation}$
* $\textbf{Experimental Validation}$: Experimental results, particularly the effect of batch normalization and bias on robustness, provide almost satisfiable empirical support for the theoretical framework.

**Weaknesses:**

* $ \textbf{Clarity and Readability}$: Some areas, such as the detailed mathematical formulations, could be streamlined or better explained.

* $\textbf{Limited Experiment Scope}$: While the paper uses popular datasets (MNIST, CIFAR-10, and FMNIST), testing on additional complex datasets could further validate the generalizability of the proposed methods.

* $\textbf{Lack of Comparisons with Baselines}$: Although robustness and stability are discussed, the paper does not thoroughly compare the proposed Minimax BNN with other advanced models or methods, such as traditional BNNs or adversarial training frameworks.

* $\textbf{Lack of Explanation of the Problem Setting}$

* $\textbf{Depth of Discussion on out of distribution Detection}$: The out-of-distribution detection section feels limited, as it mainly lists results without exploring potential reasons or implications for  out of distribution performance in depth.

* $\textbf{Not well-plished}$: The paper does not feel well-polished, and several key sections lack sufficient justification.

**Questions:**

* How does the proposed minimax BNN approach compare to adversarial training in robustness performance?

* What are the computational implications of using minimax optimization for BNNs?

* Why do certain batch normalization and bias configurations disrupt robustness, and how can this insight be utilized in network design?

**Details Of Ethics Concerns:**

The idea behind this work is quite interesting as there are still 3-4 potential remaining pages, I highly recommend rewriting the paper from beginning to end and elaborating further on certain sections, such as Section 2, to reduce ambiguity.

---

### Official Review · Reviewer_z8jH · 2024-11-03

**Soundness:** 1
**Presentation:** 1
**Contribution:** 1
**Rating:** 1
**Confidence:** 4

**Summary:**

I struggled to understand this work, my best interpretation is that instead of learning a neural network (NN) f(x, μ), where μ are the parameters, a stochastic perturbation of such NN is learned obtaining f(x, μ) + r ξ(ρ) where r is a positive scalar and ρ parametrizes the distribution ξ, which appears to be a Gaussian distribution.

**Strengths:**

I could not understand enough of the present work to asses its strenghts.

**Weaknesses:**

To put it shortly, this work is not written to a publishable standard.
The method is not presented in a clear and precise manner, the phrasing is often confusing and of poor English level.

**On the method:**

For instance, considering the supervised setting, the method is defined in (1).
It appears that the stochastic NN f(x, μ) + r ξ(ρ) is sometimes referred to as g() and sometimes as h().
Moreover, "pre()" is never defined, and the other quantities defining (1) are actually introduced only after (2), which is the representation learning setting.
The noise ξ *appears* to be Gaussianly distributed (it s not defined in Section 2, there is a mention to the Brownian Motion in the abstract, and quoting the authors "normally we use Gaussian priors throughout the paper if without specification").
Then, in Section 3, the parametrization σ = log(1 + exp(ρ)) is employed, in which case there is no difference between changing ρ and r.

**On the confusing sentences:**

> The minimax method or game theory has been used in deep learning for a long time. The most well-known work is the generative adversarial networks (GAN) Creswell et al. (2018) and Variational auto-encoding (VAE) Kingma & Welling (2013), which formulate the networks as a two-player game problem using the encoder and decoder.

I do not see how it is possible to frame a VAE as a competitive or min-max game, surely not just having an encoder and a decoder suffices...

> The contribution of this paper includes 2 points. First, the experiments of MinMax BNN verify that enough embedding dimension for CNN without bias term or Batch normalization layer for uncorrelated representation learning from a robustness perspective, and Batch normalization (BN) and bias, especially BN will have large impact on the robustness of simple CNN models trained by MNIST data. Second, a well-trained neural network can use stochastic sampling neural networks to find the suitable radius to do the out-of-distribution detection and estimate the data similarity both in representation learning and supervised learning with easier implementation and more intuitive in contrast to Bayesian Neural Networks.

I could not understand at all what was being argued here.

It should be noted that nowhere in the manuscript (aside from introduction and conclusions) do "uncorrelated representation" appears aside from the paragraph:

> The previous paper Yu et al. (2020) claims that the rate-distortion loss should have enough dimensions to make the learned features of the subspace uncorrelated. We have shown that some condition might make the learning features is isotropic to the perturbation.

Where I could not understand what the authors meant.

> There are two cases for the training process. For case 1 we will update r via golden search for every new sampling noise ξ both in maximum and minimum process, and for case 2 only update r for the minimum process.

As above.

**Moreover:**

The work appears to have been rushed in the best scenario: leftover highlighting in the abstract, leftover Contributions and Acknowledgments sections, a truncated Appendix at the end.

**Questions:**

Would it please be possible to address all the concerns raised in the Weaknesses section?

---

### Note · Authors · 2024-11-25

I have read and agree with the venue's withdrawal policy on behalf of myself and my co-authors.